# Endogenous Levels of Gamma Amino-Butyric Acid Are Correlated to Glutamic-Acid Decarboxylase Antibody Levels in Type 1 Diabetes

**DOI:** 10.3390/biomedicines10010091

**Published:** 2021-12-31

**Authors:** Henrik Hill, Andris Elksnis, Per Lundkvist, Kumari Ubhayasekera, Jonas Bergquist, Bryndis Birnir, Per-Ola Carlsson, Daniel Espes

**Affiliations:** 1Department of Women’s and Children’s Health, Uppsala University, 75185 Uppsala, Sweden; henrik.hill@kbh.uu.se; 2Department of Medical Cell Biology, Uppsala University, 75123 Uppsala, Sweden; andris.elksnis@mcb.uu.se (A.E.); Bryndis.Birnir@mcb.uu.se (B.B.); per-ola.carlsson@mcb.uu.se (P.-O.C.); 3Department of Medical Sciences, Uppsala University, 75309 Uppsala, Sweden; per-lundkvist@medsci.uu.se; 4Department of Chemistry, Analytical, BMC, Uppsala University, 75237 Uppsala, Sweden; Kumari.Ubhayasekera@kemi.uu.se (K.U.); Jonas.Bergquist@kemi.uu.se (J.B.); 5Science for Life Laboratory, Department of Medical Cell Biology, Uppsala University, 75123 Uppsala, Sweden; 6Science for Life Laboratory, Department of Medical Sciences, Uppsala University, 75309 Uppsala, Sweden

**Keywords:** type 1 diabetes, GABA, islets of Langerhans GAD-autoantibodies

## Abstract

Gamma-aminobutyric acid (GABA) is an important inhibitory neurotransmitter in the central nervous system (CNS) and outside of the CNS, found in the highest concentrations in immune cells and pancreatic beta-cells. GABA is gaining increasing interest in diabetes research due to its immune-modulatory and beta-cell stimulatory effects and is a highly interesting drug candidate for the treatment of type 1 diabetes (T1D). GABA is synthesized from glutamate by glutamic acid decarboxylase (GAD), one of the targets for autoantibodies linked to T1D. Using mass spectrometry, we have quantified the endogenous circulating levels of GABA in patients with new-onset and long-standing T1D and found that the levels are unaltered when compared to healthy controls, i.e., T1D patients do not have a deficit of systemic GABA levels. In T1D, GABA levels were negatively correlated with IL-1 beta, IL-12, and IL-15 15 and positively correlated to levels of IL-36 beta and IL-37. Interestingly, GABA levels were also correlated to the levels of GAD-autoantibodies. The unaltered levels of GABA in T1D patients suggest that the GABA secretion from beta-cells only has a minor impact on the circulating systemic levels. However, the local levels of GABA could be altered within pancreatic islets in the presence of GAD-autoantibodies.

## 1. Introduction

The amino acid gamma-aminobutyric acid (GABA) is an essential and mainly inhibitory neurotransmitter in the central nervous system (CNS). Endogenous GABA acts as a mediator in peripheral tissues and is found at its highest concentrations in immune cells [1] and pancreatic beta-cells [2]. GABA is synthesized from glutamate by the enzyme glutamic acid decarboxylase (GAD), which is present in two isoforms, GAD65 and GAD67 [3]. Autoantibodies against GAD are associated with type 1 diabetes (T1D), and a high GAD-titer is a strong predictor of rapid loss of beta-cell mass [4].

Endogenous GABA levels can be detected in blood in concentrations high enough to activate GABA-A receptors [5]; however, where it originates from and how the levels are regulated is not fully understood. It can be secreted from circulating immune cells [6,7] and pancreatic beta-cells [8]. Additionally, GABA synthesized in the brain can reach the circulation via the recently discovered glymphatic drainage system [9]. GABA exerts a predominantly suppressive effect on immune cells, inhibiting the production of inflammatory cytokines via the GABA-A receptor, which is expressed on T-cells, B-cells, and some mononuclear cells [6]. We recently found that GABA regulates the release of inflammatory cytokines from peripheral blood mononuclear cells and CD4+ T-cells. Its effects were especially potent on immune cells derived from patients with T1D [10]. Additionally, in a sub-set, we detected circulating levels of GABA in plasma that correlated to several cytokines [10].

In addition to its immune-modulatory effects, GABA has also been found to stimulate beta-cell regeneration via both beta-cell proliferation and trans-differentiation [11,12,13]. In experimental studies, GABA has even been able to reverse diabetes by restoring the beta-cell mass [14]. GABA is therefore highly interesting as a potential drug candidate for regenerative therapies in T1D. We have recently demonstrated the safety of a controlled-release GABA analog [15], and are currently conducting the first clinical proof-of-concept study investigating the potential stimulatory effects of GABA on beta-cell proliferation in patients with manifest T1D (clinicaltrials.gov ID NCT03635437).

However, little is known about how the endogenous levels of GABA are related to the disease state and metabolic features in T1D. Hence, we have investigated endogenous plasma levels of GABA in both healthy controls and patients with long-standing and new-onset T1D.

## 2. Materials and Methods

The study was approved by the Regional Research Ethical Committee in Uppsala and was conducted in accordance with the principles of the Declaration of Helsinki as revised in 2013. All participants were provided oral and written information and signed a written consent before inclusion in the study. In total, *n* = 45 healthy controls (HC), *n* = 60 patients with long-standing T1D, and *n* = 13 patients with new-onset T1D were included. 

All visits were performed in the morning after an overnight fast. Peripheral venous blood was collected for analyses at the Clinical Chemistry Laboratory, Uppsala University Hospital. According to clinical routine, patients were considered positive for autoantibodies if anti-GAD was >5 IU/mL and anti-IA2 > 7 kU/L. Plasma C-peptide was detectable if above 0.01 nmol/L using the standard clinical assay. C-peptide measurements were missing for *n* = 2 healthy controls. Plasma was collected from EDTA vials and stored at −70 °C for later analysis of GABA and cytokines. Plasma aliquots were available for cytokine analysis from *n* = 27 HC, *n* = 45 long-standing T1D, and *n* = 11 new-onset T1D patients. 

Plasma levels of GABA were analyzed by a validated protocol based on ultra-performance liquid chromatography tandem mass spectrometry. Samples were prepared by spiking with 30 µL of 100 ng/mL corresponding d6GABA into 100 µL plasma followed by liquid–liquid extraction after protein precipitation with methanol prior to the analysis. The chromatographic separation of the targeted GABA was achieved by using ultra-performance liquid chromatography (UPLC, Waters ACQUITY^®^, Milford, MA, USA) coupled with tandem mass spectrometry (XEVO^®^ TQ-S, Milford, MA, USA). The chromatographic separation of GABA was achieved on a HILIC, SS (100 × 2.1 mm, 1.8 µm; HILICON, Umeå, Sweden) at 40 °C. The gradient elution was carried out with a binary solvent system consisting of 20 mM NH4OAc in water (solvent A), acetonitrile (solvent B) at a flow rate of 0.35 mL/min. The applied linear gradient profile started with 10% A for 5 min. Then, in 8 min, the proportion of solvent A was increased to 35%. Subsequently, in 8.5 min, the proportion of solvent A was decreased to 10%. This mobile phase condition was kept for 12 min. Finally, the mobile phase was switched back to initial conditions in 0.10 min, and the column was allowed to re-equilibrate. A 5 μL aliquot of each sample was injected for analysis and the total analytical time was 12 min. 

The mass spectrometric detection was performed using electrospray ionization in the positive ionization mode (ESI+) with nitrogen and argon serving as desolvation and collision gas, respectively. The data acquisition range was 100–500 *m*/*z*. Quantification was based on a multiple reaction monitoring (MRM) method with a deuterated isotope internal standard (d6GABA). Two specific transitions were chosen, one for confirmation (the “qualifier”) and one for quantification (the “quantifier”), 104.06 > 69.06; 104.06 > 87.08 for GABA, respectively. The linearity of GABA was evaluated over a range of concentrations (1–5000 ng/mL), and correlation coefficients (*r*^2^) were 0.998. The limit of quantification (LOQ, single to noise ratio = 10) and coefficient of variation (CV) of GABA assay was 0.5 ng/mL, less than 10%, respectively. Precision was estimated by running quality control samples in five replicates on the same day, and three independent days; intra-assay CV ranged from 5.12–7.34%, while inter-assay CV was 2.46–5.73%. The recovery of the GABA assay was 85%. All data were acquired in centroid mode, analyzed, and processed using the MassLynxTM 4.2 software (Waters, Milford, MA, USA). Duplicate analyses of each sample were carried out, and the average values were reported (CV < 5%).

Circulating cytokines were analyzed with magnetic bead-based Luminex using two commercially available assays (cat. no. HTH17MAG-14K and cat. no. HCYP4MAG64K) from Merck Millipore (Burlington, MA, USA) according to the manufacturer’s protocol. A complete list of included cytokines is provided in Appendix A. The analyses were performed at the Plasma Profiling Unit, SciLifeLab (Stockholm, Sweden). Of the analyzed cytokines, IL-22, IL-24, IL-34, and IL-35 were excluded from analysis since >50% of the samples were below the detection level. For the remaining parameters, undetectable samples were assigned a numeric value corresponding to half of the lowest level of detection (single-value imputation).

Statistical analyses were conducted with GraphPad Prism version 9. Comparison between three groups was performed with a one-way ANOVA using Dunnet’s post hoc test for comparison with healthy controls. Comparisons between two groups were performed using an unpaired two-tailed *t*-test. Correlations were computed with the Spearman rank-order test. Data are presented as means ± SEM. *p*-values < 0.05 were considered statistically significant.

## 3. Results

Healthy controls and patients were of similar age, and the gender distribution within the groups was similar. The patients with long-standing T1D had a higher BMI when compared to healthy controls. As expected, fasting glucose levels and HbA1c were higher in the T1D patients. None of the healthy controls had GAD- or IA-2 autoantibodies, for complete descriptive characteristics, see Table 1.

The endogenous systemic levels of GABA are not decreased in T1D patients when compared to HC (Table 1). In addition, the levels are also similar in patients with new-onset T1D (Figure 1A). Among the patients with long-standing T1D, C-peptide was still detectable in *n* = 13 individuals, i.e., meaning that there are functionally remaining beta-cells. However, the GABA levels were similar in the C-peptide-positive patients compared to patients without detectable C-peptide (15.4 ± 0.75 vs. 15.6 ± 0.6; *p* = 0.85). The GABA levels were similar also when comparing patients without detectable C-peptide with healthy controls (15.6 ± 0.6 vs. 14.1 ± 0.6; *p* = 0.06).

Endogenous GABA levels were negatively correlated to age in both HC (*r* = −0.35, *p* = 0.02) and long-standing T1D patients (*r* = −0.33, *p* = 0.009). In patients with long-standing T1D, GABA levels were also negatively correlated to the age at onset (*r* = −0.36, *p* = 0.005), but not to disease duration (*p* = 0.7). Additionally, GABA levels were positively correlated to fasting glucose (*r* = 0.26, *p* = 0.04), but not HbA1c (*p* = 0.1) in long-standing T1D patients. GABA levels were not correlated to markers of metabolic control in HC (i.e., C-peptide, glucose, HbA1c, or BMI).

Interestingly, among GAD-positive T1D patients (*n* = 45), a negative correlation was observed between GAD-titers and endogenous GABA levels (Figure 1B). Although the reference limit for GAD-autoantibodies is >5 IU/mL, values between 5–50 are often clinically considered as low. There was *n* = 35 T1D patients who fulfilled a stricter criterion for GAD positivity (>50 IU/mL), and in this group, a stronger correlation to GABA levels was observed (*r* = −0.40, *p* = 0.018). In all T1D patients, GABA levels were positively correlated to IL-37 (*r* = 0.30, *p* = 0.026) and IL-36 beta (*r* = 0.34, *p* = 0.01). Additionally, in T1D patients, a negative correlation was observed between GABA levels and IL-12 (*r* = −0.29, *p* = 0.033), IL-15 (*r* = −0.29, *p* = 0.033) and IL-1 beta (*r* = −0.28, *p* = 0.034) (Figure 1C–G). The levels of circulating cytokines were similar in both HC and patients with T1D (Appendix A).

In HC, we observed no correlation between GABA levels and circulating cytokines.

## 4. Discussion

Using mass spectrometry, we found that the systemic levels of endogenous GABA are detectable in both healthy controls and T1D patients with varying disease duration. Interestingly, the circulating levels of GABA are not decreased in long-standing T1D despite the loss of beta-cells. This is in line with our previous study in a smaller cohort, based on another method (ELISA), in which we found that the GABA levels were slightly increased in T1D patients [10]. In addition, we also found that the endogenous GABA levels in patients with new-onset T1D are similar to the levels of HC. In light of recent experimental findings regarding GABAs effect on beta-cell proliferation, it is of interest to highlight that patients with T1D do not have a deficit in endogenous plasma GABA levels. Apparently, the GABA secretion from beta-cells does not impact, or only modestly, affect the systemic levels. This is apparent from the observed similar GABA levels in C-peptide-negative T1D patients (i.e., <0.01 nmol/L) and HC. However, levels of GAD autoantibodies were found to correlate negatively with plasma GABA levels. One could therefore speculate that the presence of GAD-autoantibodies restricts the production of GABA from glutamate. GAD autoantibodies could impact GABA levels by hampering an increase of production in both immune cells and remaining beta-cells. Given the potential impact of GAD-autoantibodies, and the minor systemic effect of GABA secreted from beta-cells, the local levels of endogenous GABA in pancreatic islets could likely be affected without a detectable systemic decrease. Theoretically, local pancreatic GABA levels could be detected using magnetic resonance spectroscopy (MRS), as shown in the brain [16]. However, due to technical limitations related to disturbances from breathing motions, this technique would be difficult to apply for quantification within the pancreas. Even if successful, the resolution would not allow for the detection of pancreatic islets.

Endogenous GABA levels were negatively correlated with age in both HC and patients with T1D. The source of circulating GABA cannot be discriminated, but given that beta-cells seems to have a minor effect on systemic levels and that the beta-cell mass is maintained rather stable over the lifespan in healthy individuals [17], the decline is most likely not related to the beta-cell mass. However, the decline could be related to the age-related loss of gray matter in the brain, which has been observed to result in decreased levels of GABA in the CNS using MRS [18]. This in turn could affect the systemic GABA levels via the glymphatic drainage system.

We found that the systemic levels of GABA correlated with circulating cytokines in patients with T1D, but not in HC. This highlights GABAs complex role in immune regulation and our previous findings that the effects of GABA on immune cells differ in T1D when compared to immune cells from HC [10]. Of the cytokines found to correlate with GABA levels, IL-1 beta, IL-36, and IL-37, are members of the IL-1 family, which is central to regulating the inflammatory response. Of the 11 known members of the IL-1 family, we also analyzed IL-33 and IL-38, but they were not correlated to GABA levels.

We found that GABA levels positively correlated to IL-36 beta and IL-37 levels. IL-36 beta is mainly secreted from monocytes and B-cells, which by activating NF-kappa-B and MAPK triggers a pro-inflammatory response. Interleukin-37, however, exerts anti-inflammatory effects by suppressing NF-kappa-beta and MAPK activation [19], i.e., an opposite effect to IL-36 beta.

GABA levels were found to negatively correlate with IL-1 beta, IL-12, and IL-15, all pro-inflammatory cytokines. IL-1 beta is considered a key mediator of beta-cell destruction. However, only a low number of islet cells have been found to be positive for IL-1 beta [20], and clinical trials with IL-1 receptor antagonists have failed to show clinical efficacy in T1D [21]. Interleukin-12 is mainly produced by activated monocytes and macrophages, and by binding to its receptor, it activates STAT4, which is essential for maintaining Th1 effector cells [22]. Interleukin-15 is elevated in T1D patients [23], and IL-15-blockade in non-obese diabetic (NOD) mice can reverse manifest disease [24]. Interestingly, we have previously found that the circulating levels of IL-15 are lower in patients with long-standing T1D, which maintain C-peptide despite >10 years of disease duration [25]. Taken together, we found that in T1D patients, GABA levels negatively correlate with several pro-inflammatory cytokines. Considering that the local levels of GABA in pancreatic islets may impact resident- and circulating immune cells, this may be of great importance for regulating the immune response within islets. Hence, a decline in beta-cell mass may further enhance an immune mediated response due to the loss of local GABA secretion.

A limitation of the current study is that the source of secreted endogenous GABA was not discriminated. Thus meaning, although the systemic levels are similar in both HC and patients with T1D the local concentration and secretion of GABA in different tissues could still be influenced. Perhaps of most importance the local GABA levels within pancreatic islets are most likely affected in T1D due to the loss of beta-cells.

## 5. Conclusions

The systemic endogenous GABA levels are not altered in patients with long-standing or new-onset T1D, but are negatively correlated with levels of GAD autoantibodies.

## Figures and Tables

**Figure 1 biomedicines-10-00091-f001:**
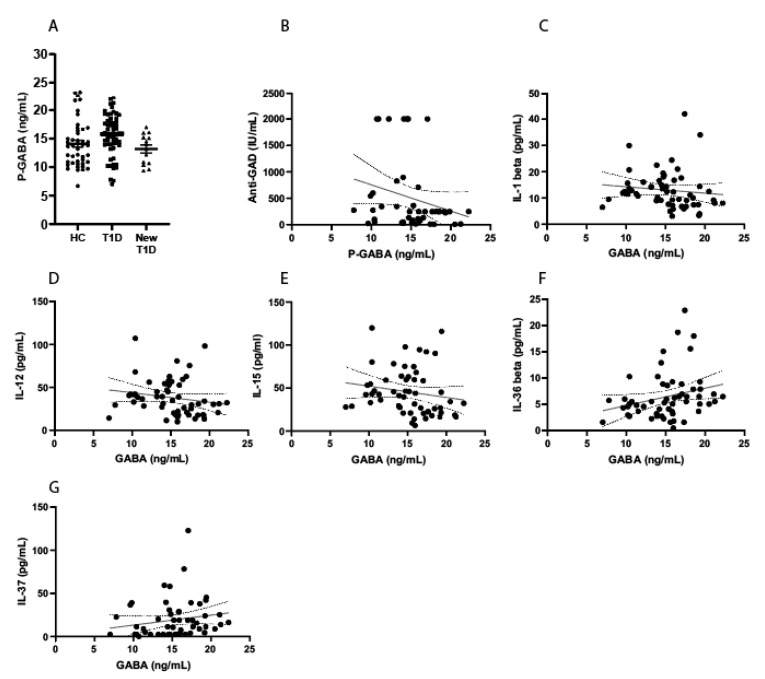
Circulating endogenous GABA levels in healthy controls and patients with type 1 diabetes. (**A**) The systemic endogenous levels of GABA were detected using mass spectrometry in healthy controls (HC, *n* = 45), patients with long-standing type 1 diabetes (T1D, *n* = 60), and in patients with new-onset T1D (New T1D, *n* = 13). No difference in circulating GABA levels was observed in T1D patients when compared to healthy controls. Values are presented as mean ± SEM. (**B**) Among the patients with T1D, *n* = 45 were positive for GAD-autoantibodies and among these patients, a negative correlation with GABA levels was observed (*r* = −0.3, *p* = 0.03). Note that the upper limit for the clinically used GAD autoantibody assay is 2000 IU/mL. (**C**–**F**) Correlations with the circulating endogenous GABA levels and circulating cytokines were computed in T1D patients (*n* = 56) and were found to be negatively correlated with circulating levels of IL-1 beta (**C**), IL-12 (**D**) and IL-15 (**E**), whereas the GABA levels were positively correlated with IL-36 beta (**F**) and IL-37 (**G**).

**Table 1 biomedicines-10-00091-t001:** Descriptive data of healthy controls and patients with type 1 diabetes.

Parameter	HC (*n* = 45)	T1D (*n* = 60)	New-Onset T1D (*n* = 13)
Female (*n*, (%))	24 (53%)	28 (47%)	6 (46%)
Age (years)	29.8 ± 1.6	28.4 ± 0.8	24.2 ± 0.7
Disease duration (years)	n/a	16.3 ± 0.8	0.17 ± 0.03
Age at onset (years)	n/a	12.1 ± 0.9	24.2 ± 0.7
BMI (kg/m^2^)	23.1 ± 0.4	24.7 ± 0.5 *	22 ± 0.6
fP-Glucose (mmol/L)	5.3 ± 0.07	11.6 ± 0.6 ***	8.1 ± 1.0 *
HbA1c (mmol/mol)	31.2 ± 0.4	61.9 ± 1.7 ***	69.2 ± 7.4 ***
Detectable C-peptide (*n*, %)	43 (100%)	13 (22%)	13 (100%)
C-peptide (nmol/L)	0.61 ± 0.03	0.08 ± 0.03 ***	0.3 ± 0.03 ***
GABA (ng/mL)	14.1 ± 0.6	15.6 ± 0.5	13.3 ± 0.7
GAD positive (*n*, %)	0 (0%)	34 (57%)	10 (77%)
IA-2 positive (*n*, %)	0 (0%)	31 (52%)	10 (77%)

Descriptive clinical data for healthy controls (HC) and individuals with long-standing type 1 diabetes (T1D). Plasma C-peptide was analyzed according to clinical routine and was detectable if above 0.01 nmol/L. C-peptide concentrations were missing for *n* = 2 HC and are only reported for long-standing T1D patients with detectable levels (*n* = 13). Patients were considered positive for autoantibodies if anti-GAD was >5 IU/mL and anti-IA2 > 7 kU/L (according to clinical routine). Statistical comparisons based on one-way ANOVA using Dunnet’s test based on comparisons with HC. * denotes *p* < 0.05 and *** *p* < 0.001. All values are given as mean ± SEM. Abbreviations: BMI, body mass index; fP, fasting plasma; HbA1c, glycated hemoglobin; GAD, glutamic acid decarboxylase; IA2, tyrosine phosphatase-like protein islet antigen-2.

## Data Availability

Data will be made available upon reasonable request.

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
