# Peer review of "Endogenous Levels of Gamma Amino-Butyric Acid Are Correlated to Glutamic-Acid Decarboxylase Antibody Levels in Type 1 Diabetes"

_biomedicines, 2021, doi:10.3390/biomedicines10010091_

Round 1

Reviewer 1 Report

Subject Appropriateness of the Manuscript:

The topic of this manuscript falls within the scope of Biomedicines

Recommendation

Major revision

Comments

In the manuscript entitled: “Endogenous levels of gamma amino-butyric acid are correlated to glutamic-acid decarboxylase antibody levels in type 1 diabetes” the Authors aimed to investigate the relationship between circulating GABA and disease state and metabolic features in T1DM. For this purpose they quantified plasma level of GABA, pro-inflammatory cytokines as well as clinical parameters (fasting glucose, HbA1c, C-peptide, GAD- and IA-2- autoantibodies) in healthy controls and patients with long-standing  type 1 diabetes (n = 60), and in patients with new-onset  type 1 diabetes (n=13). The Authors found that patients with type 1 diabetes do not have a deficit of circulating GABA level. However, plasma GABA level negatively correlated to the pro-inflammatory cytokines such as IL-1 beta, IL-12 and IL-15 and positively correlated to Il-37 and Il-36 beta. GABA levels were also correlated to the levels of GAD-autoantibodies. This is an interesting study on the disturbance of GABA metabolism in type 1 diabetes.

The manuscript, however, has some limitations, the most important of which are:

  1. a small number of new-onset type 1 diabetic patients were enrolled into the analysis. Thus, the obtained results might be out of proportion to the data reported. How were the numbers of study samples determined, i.e. how was the study powered?

Moreover, the selection of patients for this group is questionable (23% of subjects with new -onset T1D did not have GAD or IA-2 autoantibodies, and the level of C-peptide fell only by half compared to the control)

  1. supplementary Table 1 – Analyzed circulating cytokines should be supplemented with the obtained concentrations of the analyzed cytokines. The obtained concentration values for plasma GABA would also be needed
  2. the authors did not explain the obtained relationships between plasma GABA level and age, or fasting glucose in subjects with type 1 diabetes
  3. the limitations of the study should be described at the end of the paper
  4. not all obtained results are described in the abstract

Author Response

Comments

In the manuscript entitled: “Endogenous levels of gamma amino-butyric acid are correlated to glutamic-acid decarboxylase antibody levels in type 1 diabetes” the Authors aimed to investigate the relationship between circulating GABA and disease state and metabolic features in T1DM. For this purpose they quantified plasma level of GABA, pro-inflammatory cytokines as well as clinical parameters (fasting glucose, HbA1c, C-peptide, GAD- and IA-2- autoantibodies) in healthy controls and patients with long-standing  type 1 diabetes (n = 60), and in patients with new-onset  type 1 diabetes (n=13). The Authors found that patients with type 1 diabetes do not have a deficit of circulating GABA level. However, plasma GABA level negatively correlated to the pro-inflammatory cytokines such as IL-1 beta, IL-12 and IL-15 and positively correlated to Il-37 and Il-36 beta. GABA levels were also correlated to the levels of GAD-autoantibodies. This is an interesting study on the disturbance of GABA metabolism in type 1 diabetes.

The manuscript, however, has some limitations, the most important of which are:

  1. a small number of new-onset type 1 diabetic patients were enrolled into the analysis. Thus, the obtained results might be out of proportion to the data reported. How were the numbers of study samples determined, i.e. how was the study powered?

Moreover, the selection of patients for this group is questionable (23% of subjects with new -onset T1D did not have GAD or IA-2 autoantibodies, and the level of C-peptide fell only by half compared to the control)

Reply: Among the patients with new-onset T1D there were only two individuals (15%) who did not have GAD or IA-2 antibodies which is often observed among adult patients with new-onset T1D. C-peptide levels are in line with what is expected in new-onset T1D from a clinical perspective especially when considering that the samples were acquired after initiation of exogenous insulin therapy. C-peptide levels are often observed to be more suppressed directly at diagnosis and increase after the relieve of exogenous insulin therapy and improved metabolic control.
Given that the applied method has not been used previously for measuring GABA levels in patients with T1D we lacked the necessary data for performing a proper power analysis when designing the study. Based on our previous study (Bandhage A. et al, EBioMedicine 2018) in a smaller cohort (n=17 HC, n=22 T1D) using a different method (ELISA), we decided to more than double the number of participants. From our current study a power analysis can be performed. Based on the comparison of long-standing- and new-onset T1D patients, if assuming n=13 in both groups, the power of the test is 84% (two-sided test). 

  1. supplementary Table 1 – Analyzed circulating cytokines should be supplemented with the obtained concentrations of the analyzed cytokines. The obtained concentration values for plasma GABA would also be needed

Reply: Thank you for this comment, we have now included the data on circulating cytokine levels in Supplementary Table 1 which is presented as a comparison between HC and T1D. GABA levels have now been included in Table 1.

  1. the authors did not explain the obtained relationships between plasma GABA level and age, or fasting glucose in subjects with type 1 diabetes

Reply: We have now included a section in the discussion (page 6, line 195-202) which highlights these finding. Especially the negative correlation with systemic levels of GABA and age is of interest since it has been demonstrated with magnetic resonance spectroscopy (MRS) that the regional concentrations of GABA in the brain is lower in older subjects which may explain the negative correlation we observed (Maes C. et al, Hum Brain Mapp 2018). As for glucose one could speculate that it is related to the glucose effects on immune cells and that this in turn could affect the secretion of GABA but the exact mechanisms for how GABA secretion is regulated from immune cells is to the best of our understanding poorly understood.    

  1. the limitations of the study should be described at the end of the paper

Reply: Thank you for this comment, we have now included a section which describes the limitations of the paper (page 7, line 231-235).

  1. not all obtained results are described in the abstract

Reply: Thank you for this comment, we have now included all data on correlations with circulating cytokines in the abstract as this is part of the main findings. Due to the word limitation of the abstract we have however not been able to describe all results within the abstract.

Reviewer 2 Report

The manuscript by Hill et al explores the role of circulating GABA in the progression of T1D. This is a well -designed study which provides some useful information on the correlation between GABA and inflammatory cytokines. The paper is also well-written and the interpretation of results is sound and of scientific merit.

1. However, the presentation of the results can be improved upon. For instance in the results section, the authors mention several correlations in their text pertaining to GABA levels and several cytokines in T1D individuals. But none of this data is shown in the figure. Only two panels are presented here. Why not present all the data (correlation with different cytokines) in several different panels?

2. Is it not expected that there would be a negative correlation between GABA levels and GAD autoantibodies as GAD is the enzyme that  is involved in GABA synthesis, and if autoantibodies are present against this, then there would most likely result in lowered levels of GABA.

3. Which cytokines had their levels altered between T1D and HC? And how were the correlation coefficients with GABA for these specific ones?

Author Response

The manuscript by Hill et al explores the role of circulating GABA in the progression of T1D. This is a well -designed study which provides some useful information on the correlation between GABA and inflammatory cytokines. The paper is also well-written and the interpretation of results is sound and of scientific merit.

  1. However, the presentation of the results can be improved upon. For instance in the results section, the authors mention several correlations in their text pertaining to GABA levels and several cytokines in T1D individuals. But none of this data is shown in the figure. Only two panels are presented here. Why not present all the data (correlation with different cytokines) in several different panels?

Reply: Thank you for this comment, we have included figures of the correlations with cytokines in Figure 1.

  1. Is it not expected that there would be a negative correlation between GABA levels and GAD autoantibodies as GAD is the enzyme that  is involved in GABA synthesis, and if autoantibodies are present against this, then there would most likely result in lowered levels of GABA.

Reply: The levels of GABA are decreased with increasing levels of GAD autoantibodies which we, like you, believe is related to the enzymatic effect of GAD. 

  1. Which cytokines had their levels altered between T1D and HC? And how were the correlation coefficients with GABA for these specific ones?

Reply: We have now computed comparisons of cytokine levels between HC and T1D and found that all of the analyzed cytokines were at similar levels. We have included a comparison between HC and T1D regarding circulating cytokines in Supplementary Table 1.

Round 2

Reviewer 1 Report

the manuscript has been sufficiently improved to warrant publication in Biomedicines

Reviewer 2 Report

No further comments